# *Botrytis cinerea* Loss and Restoration of Virulence during In Vitro Culture Follows Flux in Global DNA Methylation

**DOI:** 10.3390/ijms23063034

**Published:** 2022-03-11

**Authors:** James Breen, Luis Alejandro Jose Mur, Anushen Sivakumaran, Aderemi Akinyemi, Michael James Wilkinson, Carlos Marcelino Rodriguez Lopez

**Affiliations:** 1Indigenous Genomics, Telethon Kids Institute, Adelaide, SA 5000, Australia; jimmy.breen@sahmri.com; 2Institute of Biological, Environmental and Rural Sciences, Edward Llywd Building, Penglais Campus, Aberystwyth SY23 3FG, UK; lum@aber.ac.uk (L.A.J.M.); ana4@aber.ac.uk (A.S.); remiakinyemi@yahoo.com (A.A.); mjw19@aber.ac.uk (M.J.W.); 3Environmental Epigenetics and Genetics Group, School of Agriculture, Food and Wine, Waite Research Precinct, University of Adelaide, PMB 1, Glen Osmond, SA 5064, Australia

**Keywords:** grey mould fungus, virulent, epigenetic, fungal pathogen culture, Whole Genome Bisulfite Sequencing, Methylation Sensitive Amplified Polymorphisims

## Abstract

Pathogenic fungi can lose virulence after protracted periods of culture, but little is known of the underlying mechanisms. Here, we present the first analysis of DNA methylation flux at a single-base resolution for the plant pathogen *B. cinerea* and identify differentially methylated genes/genomic regions associated with virulence erosion during in vitro culture. Cultures were maintained for eight months, with subcultures and virulence testing every month. Methylation-sensitive amplified polymorphisms were performed at monthly intervals to characterise global changes to the pathogen’s genome during culture and also on DNA from mycelium inoculated onto *Arabidopsis thaliana* after eight months in culture. Characterisation of culture-induced epialleles was assessed by whole-genome re-sequencing and whole-genome bisulfite sequencing. Virulence declined with time in culture and recovered after inoculation on *A. thaliana*. Variation detected by methylation-sensitive amplified polymorphisms followed virulence changes during culture. Whole-genome (bisulfite) sequencing showed marked changes in global and local methylation during culture but no significant genetic changes. We imply that virulence is a non-essential plastic character that is at least partly modified by the changing levels of DNA methylation during culture. We hypothesise that changing DNA methylation during culture may be responsible for the high virulence/low virulence transition in *B. cinerea* and speculate that this may offer fresh opportunities to control pathogen virulence.

## 1. Introduction

*Botrytis cinerea* is a pathogenic ascomycete responsible for grey mould on a diversity of plant tissue types across hundreds of dicotyledonous plant species [1,2]. It is estimated that this fungus causes annual losses of up to USD 100 billion worldwide [2]. The wide variety of tissues and species infected by *B. cinerea* suggests it is highly plastic in its ability to penetrate a host thanks to a large ‘arsenal of weapons’ at its disposal. *B. cinerea* is a capable saprotroph and necrotroph, with different genetic types often showing a trade-off between saprotrophic and necrotrophic capabilities [3]. As with other pathogens, *B. cinerea* undergoes complex transcriptional and developmental regulation to orchestrate interactions with its host. However, virulence levels of *B. cinerea* strains are not necessarily a fixed feature. For example, virulence has been observed to diminish during protracted in vitro culture [4]. Degenerated cultures have been reported in a wide range of pathogenic fungi [5,6,7], although very little is known about why cultures lose virulence. Several factors have been postulated as possible causes of virulence erosion, including dsRNA mycoviruses [8,9], loss of conditional dispensable chromosomes [10,11] or culture-induced selection of non-virulent strains. However, it is difficult to reconcile mycoviruses infection or chromosome loss as possible causes with a characteristic common to almost all in vitro-derived non-virulent fungal strains; virulence is restored after one passage on their host [5,6]. Furthermore, cultured fungal strains lose virulence irrespective of whether the parent culture originated from a single or multiple-spore colony [5], discounting the additional possibility of culture-induced selection favouring non-virulent strains.

The reversibility of this aspect of the observed phenotype comes in response to changes in the growing environment and so should be viewed as a plastic feature of the pathogen [12]. In 1942, C.H. Waddington [13] first proposed the term epigenotype to describe the interface between genotype and phenotype. Since then, a large body of research has been carried out to better understand the role of epigenetic regulatory systems in shaping the phenotype of higher organisms in fluctuating environments [7]. Epigenetic processes operate in a number of ways to alter the phenotype without altering the genetic code [14]. These include DNA methylation, histone modifications, and mRNA editing and degradation by noncoding RNAs. Such processes are intimately entwined and often work in a synergistic way to achieve changes in phenotype [15]. DNA methylation, and more specifically cytosine methylation (i.e., the incorporation of a methyl group to carbon 5 of the cytosine pyrimidine ring (5-mC)), is the most studied epigenetic mechanism. It is present across many eukaryotic phyla, including plants, mammals, birds, fish, invertebrates and fungi and provides an important source of epigenetic control for gene expression [16]. In plants and animals, DNA methylation is known to be involved in diverse processes, including transposon silencing, X-chromosome inactivation, and imprinting [17]. In fungi, 5-mC content changes during development in *Phymatotrichum omnivorum* [18] and *Magnaporthe oryzae* [19]. Global patterns in DNA methylation have also been shown to dramatically change in lichen fungi species when exposed to the algal symbiont [20]. Whole-genome bisulfite sequencing (WGBS) in Ascomycetes has revealed that this group of fungi present repeated loci silenced by methylation but also active genes methylated within exonic regions. Zemach et al. (2010) [21] reported a correlation between gene body methylation and gene expression levels in *Uncinocarpus reesii*. More recently, Jeon et al. (2015) [19] ascribed a developmental role for DNA methylation in *M. oryzae*. Recent years have seen a dramatic increase in the depth of understanding of how epigenetic control mechanisms operate during plant/pathogen interactions [22]. However, little is known of the possible role played by DNA methylation in moderating virulence in fungal plant–pathogen interactions or in its potential role in mediating culture-induced loss of virulence. Here, we explore possible links between the erosion of pathogenicity from *B. cinerea* during protracted in vitro culture and dynamics of DNA methylation across their genomes. We used methylation-sensitive amplified polymorphisms (MSAPs) [23] to provide a preliminary survey of the methylome flux associated with a gradual reduction in *B. cinerea* pathogenicity with time culture. We next sought to identify differentially methylated regions (DMRs) associated with progressive loss of pathogenicity during in vitro culture using WGBS.

## 2. Results

### 2.1. Pathogenicity Analysis of Botrytis cinerea 

Isolates from all culture time points produced lesions on *A. thaliana* leaves, but they varied in their severity (Figure 1A). There was a progressive decline in disease scores recorded over the eight-month period (i.e., T0–T8) (Figure 1B), with loss of virulence becoming significant from 3 months (T3) onwards (*t*-test *p* < 0.05) (Figure 1B). The disease scores for the T8P challenge did not differ significantly from those recorded at T0 culture time, indicating that virulence had recovered following a single passage through a plant (Figure 1B). Conversely, T8 virulence scores were significantly lower than those obtained from T0 to T5 and also than T8P (*t*-test *p* < 0.05) (Figure 1B). Infected leaves with T0 and T8P cultures did not show significant differences in fungal DNA content measured by qPCR (Figure 1C). However, both showed significantly higher levels of fungal DNA (*t*-test, *p* < 0.05) than those infected using T8 cultures (Figure 1C).

### 2.2. Analysis of Genetic and Epigenetic Variance during Culture Using MSAPs

MSAP profiles generated 74 scored loci across the 112 samples of 8 *B. cinerea* culture times used in this study (T2-T8P). Multivariate analysis revealed that the MSAP profiles of *B. cinerea* became progressively more dissimilar to the first time point analysed (T2), with increasing culture age (Figure 2). Both PCoA (Figure 2A,C) and PhiPT values (Figure 2B) showed higher levels of culture-induced variability when using *Hpa*II than when using *Msp*I. PCoA shows that samples cultivated for 3, 4, 5, 6, and 7 months occupied intermediate Eigenspace between samples cultivated for 2 and 8 months (Figure 2C). Furthermore, a partial recovery of the MSAP profile was observed on samples cultured for 8 months after one fungal generation on the host plant (T8P) (Figure 2C). Calculated PhiPT values between each time point and T2 samples show a progressive increase in MSAP profile distance with time in culture when samples were restricted with both enzymes. Analysis of molecular variance (AMOVA) showed that the calculated PhiPT values were significantly different (*p* < 0.05) between T2 and time points T6, T7, T8 and T8P when using *Msp*I and T7, T8 and T8P when using *Hpa*II (Figure 2B). Mantel test analysis showed significant correlations between disease score differences among culture times and pairwise PhiPT values from MSAP profiles generated using *Msp*I (R^2^ = 0.316; *p* = 0.005) and *Hpa*II (R^2^ = 0.462; *p* = 0.002).

### 2.3. B. cinerea Genome Re-sequencing

Divergence in MSAP profiles may occur through changes in methylation or through genetic mutation. However, the recovery of profile affinity towards that of the original inoculum after a single passage on Arabidopsis is difficult to reconcile genetically, suggesting the majority of the changes to the MSAP profiles was caused by differences in DNA methylation rather than sequence mutations. We, therefore, sought to better characterise mutational changes using a genome-wide sequencing approach. DNA extractions from two time points (1 month (T1) and 8 months (T8) in culture) were compared to the Broad Institute’s *B. cinerea* B05.10 reference genome sequence and used to assess mutational change during culture. Both samples were sequenced to an average depth of 37.47× (35.64× for the 1-month culture and 39.30× for the 8-month culture), with an average of 80% of reference bases being sequenced to a depth greater than 10×. After filtering, we found 2331 sequence variants after 8 months in culture, of which 1030 (44%) were small insertions and deletions (INDELs) and 1301 (56%) SNPs. Of these, just 454 were located within genes, including: 251 synonymous variants, 198 non-synonymous mutations (193 causing missense variations and 5 causing premature stop codons (non-sense mutations). An additional 5 variants that altered predicted splice junctions were also identified.

We next focused the search for variants within the sequence of 1577 *B. cinerea* genes with known function, including: secondary metabolism, conidiation, sclerotium formation, mating and fruit body development, apoptosis, housekeeping, signalling pathways [25] and virulence sensu lato genes. The virulence sensu lato genes included: appressorium associated genes [25], virulence sensu stricto genes [26] and plant cell wall disassembly genes (CAZyme genes) [27]. We found 68/1577 (4.3%) of the tested genes contained one or more variants between T1 and T8 (see Table 1 and Appendix A for a comprehensive list of genes with variants).

### 2.4. Characterisation of DNA Methylation Changes by WGBS

We next characterised genome-wide DNA methylation flux by conducting WGBS-triplicated genomic DNA extractions from mycelia of two different culture ages (T1 and T8) and eight-month cultures after inoculation onto *A. thaliana* plants (T8P). This yielded 187.5 million reads, ranging from 12.61 to 34.05 Gbp of sequence per sample after quality filtering. The mapping efficiency of each replicate ranged from 54.3 to 67.6%, resulting in samples that generated between 33 and 55x coverage of the 42.66 Mbp genome (Appendix A). This is the highest genome coverage achieved to date for any fungal species after bisulfite sequencing. For each sample, we covered over 91–93% of all cytosines in the genome (Appendix A), with all samples having at least 81% of cytosines covered by at least 4 sequencing reads, allowing methylation level of individual sites to be estimated with reasonable confidence.

WGBS identified an average of 15,716,603 mC per sample, indicating an average methylation level of 0.6% across the genome as a whole (Appendix A). The most heavily methylated context was CHH, followed by CG and CHG (where H is A, C or T) (Appendix A). Global levels of mC did not significantly change with culture time, although methylation in the rarer CG and CHG contexts increased significantly (*t*-test, *p* = 0.0008 and 0.0018) between 1 and 8 months in culture (Figure 3A). However, modest declines were apparent in both contexts on eight-month-old cultures inoculated onto *A. thaliana* (T8P) (Figure 3A); they proved not significant (*t*-test, *p* > 0.05). Analysis of local levels of DNA methylation across the largest *B. cinerea* contig (Supercontig 1.1) showed that DNA methylation was unevenly distributed and locally clustered (Figure 3B–D). Observed clustering patterns were similar in all analysed samples (Figure 3B–D).

We next investigated the fine-scale distribution of DNA methylation across genic and regulatory regions by surveying the location and density of mCs on all *B. cinerea* genes (exons and introns) and presumed promoter regions, defined here as 1.5 kb upstream of the transcription starting site (TSS). Among these, mC levels increased between 1000 and 500 bp upstream of the TSS. This was followed by a sharp decrease and then an increase in methylation before and after the start of the coding sequence, respectively (Figure 4). When time points T1, T8 and T8P were compared in this genomic context, methylation levels were higher on T8P samples (Figure 4), in parallel with that observed at a whole-genome level (Figure 3A).

The same methylation density analysis was then carried out for five housekeeping genes in botrytis (i.e., G3PDH (BC1G_09523.1); HSP60 (BC1G_09341.1); actin (BC1G_08198.1 and BC1G_01381.1) and beta tubulin (BC1G_00122.1). All five loci possessed low levels of DNA methylation in every context, and there were no changes in DNA methylation between culture time points (i.e., T1, T8 and T8P) (data not shown).

In common with most genes, those encoding putative CAZymes, proteins secreted by *B. cinerea* upon plant infection (Appendix A) [27], contained more methylation in regions immediately upstream of the TSS (Figure 5A). However, these genes showed higher levels of methylation after 8 months in culture than at T1 and T8P. The observed increase in methylation on T8 samples was primarily driven by increased methylation in the CHG and CG (Figure 5B,C) contexts. In contrast, CHH sites (Figure 5D) showed the highest levels of methylation on the T8P samples, following the general trend observed both at a whole-genome level (Figure 3A) and by all *B. cinerea* genes (Figure 4).

### 2.5. Detection of Culture-Induced DMRs

We sought to identify DMRs between three culture times (T1, T8 and T8P) by comparing methylation levels across the whole genome of all samples implementing swDMR sliding window analysis. Analysis of DMR length distribution showed DMR sizes ranging from 12 to 4994 bp (Figure 6A). The sliding window approach identified 2822 regions as being significantly differentially methylated in one of the samples compared to the other two for all mCs (Table 2 and Appendix A). Overall methylation levels of DMRs decreased in all contexts (CG, CHG and CHH) as time in culture progressed (from T1 to T8), but this was followed by a recovery of DNA methylation levels after 8-month cultures were inoculated onto *A. thaliana* (T8P) (Figure 6B). However, it is worth noting that T8 showed a larger number of outlier DMRs (greater than two standard deviations away from the mean) that exhibited significantly higher levels of methylation than the average (Figure 6B).

When examined individually, variance in DMRs between samples comprised of two main pattern types (Table 2 and Appendix A): 1. 57.3% of the detected DMRs showed a recovery pattern inoculation on *A. thaliana* such that there was no difference between T1 and T8P samples, but methylation levels diverged significantly in T8 (T1 = T8P< >T8 (FDR < 0.01)). 2. The remaining DMRs showed a non-recovery pattern (i.e., T1< >T8P (FDR < 0.01)). Two subtypes were found for DMRs showing a DNA methylation recovery pattern: 1. DMRs showing an increase in methylation with time in culture (T1 = T8P < T8 (26.82%) (Type 0 hereafter) and 2. DMRs showing a decrease in methylation level with time in culture (T1 = T8P > T8 (30.47%)) (Type 2). Equally, non-recovery DMRs can be divided into two categories: 1. DMRs showing a decrease in methylation level with time in culture and no change in methylation level following inoculation on *A. thaliana* (T1 > T8 = T8P (16.02%)) (Type 1a) and 2. DMRs not showing changes in methylation level during culture but an increase in methylation after inoculation on *A. thaliana* (T1 = T8 < T8P (26.68%)) (Type 1b). Curiously, no DMRs were observed to show a progressive increase in methylation level with time in culture, and there was no change in methylation level after inoculation on *A. thaliana* (T1 < T8 = T8P).

The vast majority (84.5%) of detected DMRs overlapped with 3055 genic regions in the *B. cinerea* genome (Table 3), while 438 (15.5%) were mapped to intergenic regions. Almost all of the genes implicated (98%) included DMRs within the gene body itself, with a small majority (53.9%) also overlapping with the promoter region (Table 3 and Appendix A). The same analyses were carried out to detect DMRs for CG, CHG and CHH contexts, identifying 70, 82 and 1248 DMRs, respectively, for each context (Table 3 and Appendix A). Of these, 91.4% (CG), 89.0% (CHG) and 85.2% (CHH) overlapped with 68, 84 and 1339 genes, respectively (Table 3 and Appendix A). Finally, we conducted a search for DMRs overlapping with 1577 *B. cinerea* genes with known targeted functions that were carried out in the re-sequencing section above. Of these, 478 genes (30.3%) overlapped with one or more detected DMRs (see Table 1 and Appendix A for a comprehensive list of genes overlapping with DMRs).

## 3. Discussion

### 3.1. Culture-Induced Changes to MSAP Profiles and Virulence Are Simultaneous and Reversible

In accordance with previous reports [6,7,8,9], the virulence of *B. cinerea* cultures progressively decreased with culture age but recovered after one passage of in vivo infection on *A. thaliana* [5]. Concurrent with these changes, MSAP profiles showed a similar progressive increase in deviation from the starter culture profiles consistent with previous reports of accumulative genetic/methylome change for other species when similarly exposed to prolonged periods of culture [28]. The progressive divergence in methylation profiles as culture continued showed a positive linear correlation with the observed changes in virulence. This accords with the reported high levels of somaclonal variability arising during in vitro growth of phytopathogenic fungi, which seemingly depresses the level of virulence of the culture isolates [29]. It should be noted that the source of the increased variation during protracted culture could have either a genetic or epigenetic basis since both mutation and methylation have the capacity to perturb MSAP profiles and virulence. However, the reversal of virulence phenotype after a single passage of the cultured fungus on *A. thaliana* (T8P) strongly implies that a plastic epigenetic mechanism is primarily responsible for driving the observed decline in virulence. This is most easily explained if culture alters the global methylation profile of the genome and this, in turn, perturbs the expression control of genes responsible for virulence. For this hypothesis to hold, a particularly strong link should exist between changes in the methylation status of genes known to be implicated in virulence and the disease symptoms evoked by cultured isolates. Establishing such links requires a more precise description of both methylome flux and mutation in culture.

### 3.2. Sequence Variants Do Not Explain Loss of Virulence during Culture

The strong body of work demonstrating causal links between gene mutation and virulence in *B. cinerea* [30,31,32] means that the potential for mutation as the sole explanation of the erosion of virulence in culture warrants attention. However, the large number of mutations identified in our study contrasts with previous reports of fungal genomic stability during culture. Kohn et al. (2008) reported the great stability of the fungal genome even after 400 days of in vitro growth [33]. One explanation for the observed differences lies in the large differences in resolution and genome coverage of both approaches. As an example, in the case of the AFLP loci analysed by Kohn et al., assuming an average size of 300 bp per locus, that would mean that the cumulative number of bases included in their analysis was 6600 per sample compared to the 33,600,000 bases that would have been sequenced to a depth of 10× or more in our results. Additionally, the codominant nature of AFLP markers results in the possibility of false negatives (i.e., mutations not been identified) unless AFLP band intensity is analysed instead of band presence/absence alone (which was not the case here). Additionally, the use of 1.2% agarose gels by Kohn et al. means that SNPs and small and midsize indels (1–200 bp) could have been missed (1.2% agarose gels are optimised to discriminate DNA fragments 400–7000 bp with a minimum difference in size between PCR products of more than 200 bp). Our study, nevertheless, uncovered just 6 of 1184 genes associated with virulence (0.5%), which included a sequence variant that could conceivably affect the virulence phenotype. All six were plant cell wall disassembly genes, namely, CAZyme genes [34]. Whilst some of the loss of virulence may have been causally linked to the observed progressive erosion of virulence in culture, the subsequent restoration of virulence in vivo would also require the retention of all wild type versions of all genes to facilitate. This seems implausible as the only explanation, despite previous reports that the species is capable of sustaining high levels of standing diversity [35]. Furthermore, there is extensive functional redundancy among the 275 CAZyme genes present in the Botrytis genome [27], and it is highly unlikely that the gradual loss of one or a small number of these genes would have such a pronounced effect on the disease phenotype. Equally, it is difficult to reconcile the stochastic and infrequent nature of mutation against the smooth and progressive erosion of the virulence and subsequent restoration seen across all replicates. Thus, we are inclined not to concur with the view that genetic mutation is the primary cause of the erosion of virulence during in vitro culture of *B. cinerea* [36].

### 3.3. In Vitro Culture Affects Whole-Genome Methylation Patterns

In the present study, methylome sequencing of nine cultures across three time points revealed similar low levels of DNA methylation (0.6%) compared to that reported in other fungal species [21]. DNA methylation was not evenly distributed but clustered in a mosaic pattern, with higher levels of methylation in regions with low gene density. Jeon et al. (2015) [19] noted similar methylation clustering patterns in pathogenic fungi around gene-poor regions, rich in transposable elements, and found these to dynamically follow fungal development. However, our clustering patterns remained similar across all time points, implying TE-rich regions are unlikely to be associated with the progressive erosion of virulence in culture. Rather, the consistent accumulation of methylation within CG and CHG motifs over 8 months in culture, followed by a complete reversal (demethylation) after inoculation on *A. thaliana,* indicates that CG and CHG methylation accumulates in culture but is lost after contact with the host. This is at least consistent with a functional role for methylation in the observed loss and then restoration of virulence.

Comparative analysis between culture time points showed that when all *B. cinerea* genes were analysed collectively, there were no gross changes in DNA methylation with time in culture. However, a higher-resolution examination of the data provided greater insight. Evidence from other systems suggests that, in general, gene body methylation enhances gene expression [19,21], whereas promoter methylation represses expression [19]. In this context, our observation of an increase in methylation approximately 800 bp upstream of the transcription start site (TSS), followed by a sharp decrease in the exonic regions, is suggestive of widespread methylation-mediated gene silencing in culture, in much the same way as was shown by Jeon et al. (2015) in *M. oryza* mycelia [19]. However, our further finding of a second increase in methylation density after the TSS in several loci shows that this trend is not universal and implies that at least some methylation occurs in the body of *B. cinerea* genes, potentially leading to enhanced expression as this form of methylation accumulates in culture.

Finer level interrogation of the data suggested CAZyme genes may be one group of possible candidates that link changes in methylation during culture to functional changes in virulence. Methylation in the promoter regions of CAZyme genes negatively correlated with virulence throughout the experiment and was particularly associated with a methylation increase in the CHG and CG contexts. Conversely, global methylation levels within CAZyme gene bodies were positively correlated with virulence levels, primarily in the CHH context. As outlined above, both of these categories of methylation change are commonly associated with expressional control [19,21], suggesting a chronology of methylation that matches that of virulence. Functionally, the CAZyme gene family encodes proteins that breakdown, biosynthesise and modify plant cell wall components and are highly expressed during plant invasion in *B. cinerea* [27,34]. It is, therefore, a plausible candidate as a methylation-dependent gene family that could play a functional role in the virulent form of *B. cinerea*.

### 3.4. Culture Induced DMRs Are Reversible and Mirror Changes in Virulence

We identified 2822 candidate DMRs that exhibited a decrease in methylation between T1 to T8 across all contexts (CG, CHG and CHH), followed by a recovery of DNA methylation after culture on *A. thaliana* (T8P). A similar sequence of methylome flux has been reported previously in studies using methyltransferase knockout mutants [37]. These sites run contrary to the prevalent trend across the genome and may simply reflect a difference between the behaviour of clustered and dispersed sites of methylation or perhaps between functional groups of genes or cell types. Certainly, similar differences noted between global and local DNA methylation levels have been noted in previous studies on various organisms [19,38]. Furthermore, Jeon and colleagues [19] showed that fungal totipotent cells (mycelia) possess higher global methylation levels while cells determined to host penetration (appresoria) contain a higher number of genes with methylated cytosines but lower global levels of DNA methylation. More specifically, exogenous DNA demethylation of *B. cinerea* inhibits conidial germination [37]. This accords with the findings of our study, where 68.3% of the culture-induced DNA methylation changes (i.e., Type 0, 2 and 1a, which accounts for 57.3% of the total detected DMRs) showed a recovery pattern after a single round of inoculation on *A. thaliana*. This suggests that the majority of the DNA methylation changes accumulated during in vitro culture were reset to their original state after a single passage on the host. Viewed in this context, it could be argued that 26.7% of the observed changes (Type 1b) were not induced by in vitro culture at all but by exposure to the host. The extent to which these and alternate chronologies of methylation change are integral to the state of virulence of *B. cinerea* is at least partly dependent on the proportion of marks associated with genic regions.

Of the detected DMRs for all contexts, 84.5% overlapped with one or more genes suggesting that the great majority of culture-induced DNA methylation flux occurs in genic regions. It is possible that the observed overlap of individual DMRs with more than one gene might be partly attributable to the small average size of intergenic regions (778 to 958 bp) and genes (744 to 804 bp) in *B. cinerea* [25], especially since 98% of these DMRs overlapped, at least partially, with gene bodies. Taken collectively, this indicates that *B. cinerea* genes in general and gene bodies in particular are prone to environmentally induced change to their methylation status. When DMRs were defined using each DNA methylation context independently (i.e., CG, CHG and CHH), the vast majority (89.1% of all DMRs and 89.9% of those overlapping with genes) occurred in the CHH context, implying that DMRs in this context are responsible for the observed changes in most genes. Specifically, CHH DMRs were detected in the CAZyme gene family (see above) and with other genes linked with apoptosis and conidiation (40.0 and 37.5%, respectively), both processes being associated with higher levels of virulence. In contrast, housekeeping genes did not overlap with any DMRs. Whilst both biological processes have been previously shown to be affected by DNA methylation [19,39,40], 27.6% of all virulence genes analysed overlapped with DMRs, and some 77% of these showed recovery after a single passage on the host. Collectively, these findings are concord with the assertion of previous works that although multiple systems of gene regulation operate, epigenetic control plays an important role in regulating host/pathogen interactions [37,41,42]. Our study also adds to the growing body of evidence suggesting the importance of dynamic DNA methylation as a regulator of phenotypic plasticity of plant pathogens and interaction with the host [43].

Overall, our results show that the protracted culture of *B. cinerea* induces a hypomethylated, low pathogenic form adapted to the host’s absence or the abundance of nutrients in the culture media. We speculate that the observed change in global and local levels of DNA methylation during in vitro culture is part of a mechanism that confers plasticity to the *B. cinerea* genome to adapt to different environments. This adds to a growing consensus in the field. Butt et al. (2006) [5] proposed that in vitro culture-induced loss of virulence could be the reflexion of an adaptive trait selected to promote energy efficiency by turning off virulence genes in the absence of the host or in nutrient-rich environments and that such change could become maladaptive by restricting the pathogen to saprophytism [5]. While it has been argued elsewhere that environmentally induced epigenetic adaptive changes have the potential to induce evolutionary traps, leading to maladaptation [44], the complete reversibility of the reduced virulence phenotype observed suggests that this might not be the case here.

We propose that the next step towards a better understanding of the epigenetic regulation of virulence in fungi should be to confirm the causal links between the methylation changes proposed here and perturbations in virulence genes expression. This, together with the increasing availability of sequenced fungal genomes and methylomes and our ability to decipher the associations between changes in DNA methylation and virulence, will stimulate the understanding of the mechanisms involved in fungal virulence control. Moreover, the analysis of DMRs could potentially be used to predict gene function in non-model fungal species or even to predict pathogenicity. More importantly, if the epigenetic regulation of the virulent/non-virulent transition proposed here applies broadly to other pathogenic fungi, our findings will open the door to a new type of non-lethal fungicide aimed at maintaining pathogenic fungi as saprotrophic, which would reduce the appearance of resistant strains.

## 4. Materials and Methods

### 4.1. Botrytis cinerea Culture and Inoculation

Seven *Botrytis cinerea* cultures (IMI169558 isolate [45]) were initiated from a single frozen inoculum and cultured and harvested for 32 weeks, as described in Johnson et al. (2007) [46]. Briefly, *B. cinerea* was cultured on potato dextrose agar plates at 20 °C with a 12 h photoperiod in a Gallenkamp illuminated cooled incubator (Sanyo Biomedical Europe BV). After 4 weeks in culture (T0), the initial culture was subcultured to 7 plates containing fresh medium. Mycelium from each plate was subsequently subcultured every 4 weeks (1 month hereafter) into fresh medium. A mycelium sample was taken from all replicates for DNA extraction, and conidia was harvested from five replicates at every subculture for virulence analysis. Finally, after the T8 challenge, *B. cinerea* was isolated from the infected areas, cultured and immediately used to challenge *A. thaliana* plants to test virulence recovery (Appendix A) (T8P).

### 4.2. Plant Material

*A. thaliana* Col-0 seeds were obtained from the Nottingham Arabidopsis Stock Centre (NASC) and cultivated in Levington Universal compost in trays with 24-compartment inserts [47]. Plants were maintained in Conviron (Controlled Environments Ltd., Winnipeg, MB, Canada) growth rooms at 24 °C with a light intensity of 110 µmol m^−2^/s and an 8 h photoperiod for 4 weeks. For ease of treatment, plants were transferred to Polysec growth rooms (Polysec Cold Rooms Ltd., Worcester, UK), maintained under the same conditions.

### 4.3. DNA Isolation

In total, 126 *B. cinerea* genomic DNA (gDNA) extractions (comprised of 2 mycelium samples of each of the 7 plates at in vitro time point (T1-T8) and from of the last time point culture transplanted onto to *A. thaliana* (T8P)) were performed using the DNeasy 96 Plant Kit (Qiagen, Valencia, CA, USA) according to the manufacturer’s instructions. Isolated DNA was diluted in nanopure water to produce working stocks of 10 ng·µL^−1^. DNA from *B. cinerea-*inoculated *A. thaliana* was extracted from five-leaf samples at each time point using the DNeasy Mini Kit (Qiagen, Valencia, CA, USA), as above. The DNA was diluted to 1 ng × µL^−1^.

### 4.4. Scoring B. cinerea Lesion Phenotypes

For assessments of infection phenotypes, leaves from five *A. thaliana* Col-0 plants (leaf stage 7/8 as defined by Boyes et al. (2001) [48]) were inoculated on the adaxial surface with 5 µL of spore suspension collected at each subculture time (T0–T8P). Controls were inoculated with PDB. Disease lesions were assessed 3 days post-inoculation. A weighted scoring method was used to categorise lesion phenotypes [49]. High virulence symptoms (water-soaking, chlorosis, and spreading necrosis) were conferred a range of positive scores, and the resistant symptoms (necrosis limited to inoculation site) were given negative scores (Figure 1A). A weighted score was produced arithmetically from the lesion scores of replicates. Inoculated *A. thaliana* leaves at T0, T8 and T8P were collected 3 days after inoculation for estimation of in planta fungal development by quantitative real-time PCR (qPCR) (Appendix A).

### 4.5. Estimation of in Planta Fungal Growth by qPCR

Reaction mixtures for qPCRs (25 μL) were prepared by mixing 10 ng of DNA with 12.5 μL of SYBR™ Green Mastermix (Applied Biosystems, Warrington, UK) and primers (300 nM final concentration). Arabidopsis and Botrytis primers generated a 131 bp amplicon of the Shaggy-kinase-like gene and a 58 bp amplicon of the Cutinase A gene, respectively [24]. qPCRs were carried out using a Bio-Rad ABI7300 thermocycler using the following conditions: 15 min (95 °C), followed by 50 cycles of 15 s (95 °C), 30 s (58 °C) and 1 min (72 °C). This was followed by a dissociation (melting curve), according to the software procedure. Serial dilutions of pure genomic DNA from each species were used to trace a calibration curve, which was used to quantify plant and fungal DNA in each sample. Results were expressed as the CG11/iASK ratio of mock-inoculated samples.

### 4.6. MSAP Procedure

We used MSAPs, as described in Rodríguez López et al. (2012) [50], to reveal global patterns of variability that reflect the divergence in methylation and sequence mutations between *B. cinerea* samples; 14 samples per time point (T2 to T8P) were analysed (2 replicated DNA extractions per culture plate). For each individual sample, 50 ng of DNA were digested and ligated for 2 h at 37 °C using *Eco*RI (5U) and *Msp*I or *Hpa*II (1U) (New England Biolabs), 0.45 μM *Eco*RI adaptor, 4.5 μM *Hpa*II adaptor (Appendix A for all oligonucleotide sequences) and T4 DNA ligase (1U) (Sigma) in 11 μL total volume of 1X T4 DNA ligase buffer (Sigma), 1 μL of 0.5 M NaCl, supplemented with 0.5 μL at 1 mg/mL of BSA. Enzymes were then inactivated by heating to 75 °C for 15 min. Restriction/ligation was followed by two successive rounds of PCR amplification. For preselective amplification, 0.3 μL of the restriction/ligation products described above were incubated in 12.5 μL volumes containing 1X Biomix (Bioline, London, UK) with 0.05 μL of Preamp EcoRI primer and 0.25 μL Preamp HpaII/MspI (both primers at 10 uM) supplemented with 0.1 μL at 1 mg/mL of BSA. PCR conditions were 2 min at 72 °C, followed by 30 cycles of 94 °C for 30 s, 56 °C for 30 s, and 72 °C for 2 min with a final extension step of 10 min at 72 °C.

Selective PCRs were performed using 0.3 μL of preselective PCR product and the same reagents as the preselective amplification but using FAM-labelled selective primers (E2/H1). Cycling conditions for selective PCR were as follows: 2 min (94 °C), 13 cycles of 30 s (94 °C), 30 s (65 °C, decreasing by 0.7 °C each cycle), and 2 min (72 °C), followed by 24 cycles of 30 s (94 °C), 30 s (56 °C), and 2 min (72 °C), ending with 10 min (72 °C). Fluorescently labelled MSAP products were diluted 1:10 in nanopure sterile water, and 1 μL was combined with 1 μL of ROX/HiDi mix (50 μL ROX plus 1 mL of HiDiformamide, Applied Biosystems, Waltham, MA, USA). Samples were heat-denatured at 95 °C for 3–5 min and snap-cooled on ice for 2 min. Samples were fractionated on an ABI PRISM 3100 at 3 kV for 22 s and at 15 kV for 45 min.

### 4.7. Analysis of Genetic/Epigenetic Variability during Time in Culture Using MSAP

MSAP profiles were visualised using GeneMapper Software v4 (Applied Biosystems, Foster City, CA, USA). A qualitative analysis was performed, in which loci were scored as “present” (1) or “absent” (0) to create a presence/absence binary matrix. MSAP-selected loci were limited to amplicons in the size range of 80–585 bp to reduce the potential for size homoplasy [51]. Samples were grouped according to the cumulative period in culture at the time of collection. Samples collected after 2, 3, 4, 5, 6, 7 and 8 months were denoted as T2, T3, T4, T5, T6, T7, T8 and T8P, respectively, whereas those cultured for 8 months and then inoculated onto *A. thaliana* were labelled T8P.

The similarity between MSAP profiles obtained from primer combination E2/H1 and both enzymes (*Hpa*II and *Msp*I) was first visualised using principal coordinate analysis (PCoA) [52] using GenAlex (v.6.4) [53]. We then used analysis of molecular variance (AMOVA) [54] to evaluate the structure and degree of diversity induced by different times in culture. Pairwise PhiPT [55] comparisons between samples restricted with *Hpa*II or *Msp*I from each time point and the samples after the first passage (2 months in culture, T2) were used to infer their overall level of divergence with time in culture (i.e., the lower the PhiPT value between samples restricted using *Hpa*II or *Msp*I, the smaller the differentiation induced by culture and the same samples). AMOVA was subsequently calculated using GenAlex (v.6.5) to test the significance of PhiPT between populations [55], with the probability of non-differentiation (PhiPT = 0) being estimated over 9999 random permutations.

Mantel test analysis was used to estimate the correlation between the calculated pairwise molecular distances and the difference in virulence between culture time points. The level of significance was estimated over 9999 random permutations tests, as implemented in Genalex v6.5.

### 4.8. Methylation Analysis by WGBS

DNA from 9 biological replicates from culture of two culture ages (1, 8 months) and from 9 replicates of T8P were randomly selected for sequencing. Biological replicates were used to generate 3 pooled samples per culture age. Bisulphite treatment was performed independently from 50 ng of genomic DNA of each pooled sample using the EZ DNA Methylation-Gold™ Kit (Zymo Research, Irvin, CA, USA) according to the manufacturers’ instructions but adjusting the final column purification elution volume to 10 µL. Following bisulphite treatment, recovered DNA from each pool was used to estimate yield using a NanoDrop 100 spectrophotometer using the RNA setting.

Bisulphite-treated samples were then used to create a sequencing library using the EpiGnome™ Methyl-Seq Kit and Index PCR Primers (4–12) according to the manufacturer’s instructions. In order to provide a reference draft sequence for the alignment of the bisulphite-treated DNA and to detect any culture-induced genetic mutations, 10 ng of native (non-bisulphite treated) DNA extracted from cultures from time points of 1 month (T1) and 8 months (T8) was sequenced and compared to the reference *B. cinerea* B05.10 genome sequence. Libraries were prepared using the EpiGnome™ Methyl-Seq Kit and Index PCR Primers (Epicentre) (1–2) according to the manufacturer’s instructions.

Library yield was determined by Qubit dsDNA High Sensitivity Assay Kit. The Agilent 2100 Bioanalyzer High-Sensitivity DNA Chip was used to assess library quality and determine average insert size. Libraries were then pooled and sequenced using Illumina HiSeq2000 (Illumina Inc., San Diego, CA, USA) 100 bp paired-end V3 chemistry. The data are available at the Sequence Read Archive (PRJEB14930).

### 4.9. Sequence Analysis and Differential Methylation Analysis

Sequencing reads were trimmed to remove adaptors using *TrimGalore*! (http://www.bioinformatics.babraham.ac.uk/projects/trim_galore, accessed on 1 December 2021) and *Cutadapt* [56]. Whole-genome re-sequencing reads were aligned to the published genome from *B. cinerea* B05.10 (Broad Institute’s *B. cinerea* Sequencing Project) using *bowtie2* [57], and variants were called using *freebayes* [58] after filtering for coverage greater than 10× and >30 variant quality and removal of multi-allelic variants, variants with missing data in one sample and variants with an observed allele frequency less than 0.5 [31]. Variant categories were analysed using *CooVar* [59], *bedtools* [60] and custom scripts. Bisulfite-treated libraries were mapped using Bismark [61] and bowtie2, duplicates removed and methylation calls extracted using samtools [62] and in-house scripts (available upon request). Bisulfite sequencing efficiency was calculated by aligning reads to the *B. cinerea* mitochondrial genome scaffold (B05.10) and identifying non-bisulfite converted bases. Differentially methylated regions were called using a sliding window approach described in swDMR (https://code.google.com/p/swdmr/, accessed on 1 December 2021). The significance of the observed DMRs was determined using a three-sample Kruskal-Wallis test between T1, T8 and T8P.

## Figures and Tables

**Figure 1 ijms-23-03034-f001:**
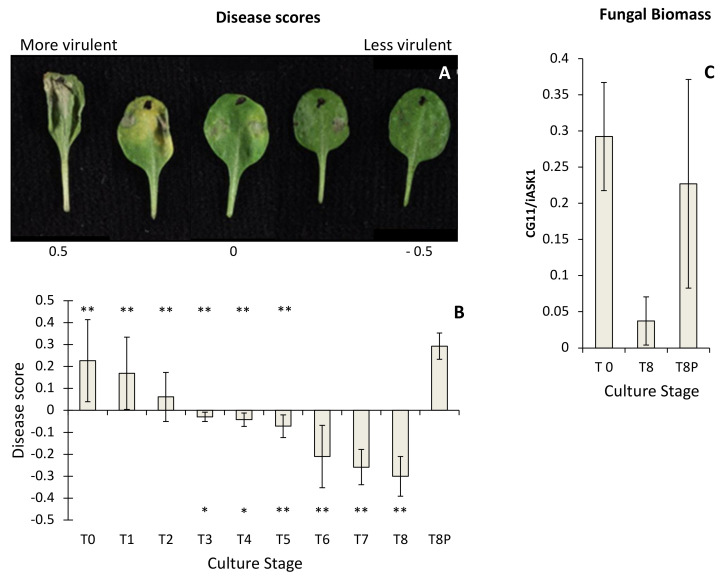
Estimation of *Botrytis cinerea* virulence on *Arabidopsis thaliana*. (**A**) Examples of *A. thaliana* Col-0 leaves drop-inoculated with *B. cinerea* and presenting disease scores ranging from −0.5 (low virulence) to 0.5 (high virulence). (**B**) Estimated *B. cinerea* virulence at each time culture point. Virulence of *B. cinerea* cultures was estimated over a period of eight months in culture (T0, initial inoculum; T8, eight months in culture) and after 8 months in culture and a single passage on *A. thaliana* (T8P). A weighted scoring method was used to categorise *B. cinerea* lesion phenotypes 3 days post-inoculation. Virulence symptoms (water-soaking, chlorosis, and spreading necrosis) were conferred a range of positive scores, and the resistant symptoms (necrosis limited to inoculation site) were given negative scores. Asterisk symbols under the horizontal axis indicate significant differences (* (*t*-test; *p* < 0.05) and ** (*t*-test; *p* < 0.01)) between T0 and the time point over the asterisk. Asterisk symbols over the horizontal axis indicate significant differences (** *t*-test; *p* < 0.01) between T8P and the time point under the asterisk. (**C**) Detection of *in planta B. cinerea* hyphal mass in *A. thaliana* Col-0 by qPCR, as described by Gachon and Saindrenan, 2004 [24].

**Figure 2 ijms-23-03034-f002:**
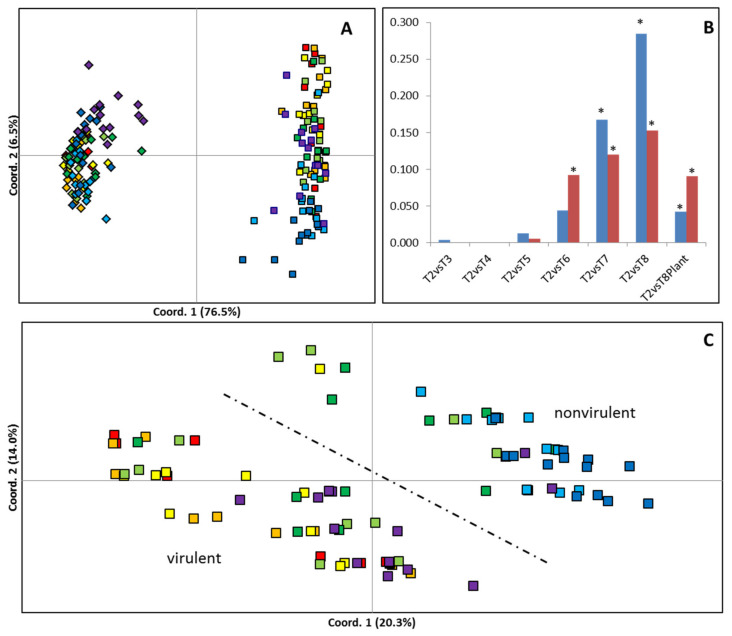
Effect of time in culture on genetic/epigenetic instability. (**A**,C) Principal coordinate diagrams based on the Euclidian analysis of methylation-sensitive amplified polymorphisms (MSAPs) using (**A**) enzymes *Hpa*II (squares) and *Msp*I (rhomboids) and using (**C**) enzyme *Hpa*II distances; 14 replicates from each time point are represented as red (T2: 2 months in culture), orange (T3: 3 months in culture), yellow (T4: 4 months in culture), light green (T5: 5 months in culture), dark green (T6: 6 months in culture), light blue (T7: 7 months in culture), dark blue (T8: 8 months in culture), and purple (T8P: 8 months + plant). The dashed line separates samples with higher average levels of virulence from those with lower average levels of virulence. (**B**) Calculated pairwise PhiPT (Michalakis and Excoffier, 1996) comparisons between samples restricted with *Hpa*II (Blue) or *Msp*I (Red) from each time point and the samples after the second passage (2 months in culture). * Indicates significantly different PhiPT values between T2 and the time point under the asterisk based on 10,000 permutations (*p* = 0.05).

**Figure 3 ijms-23-03034-f003:**
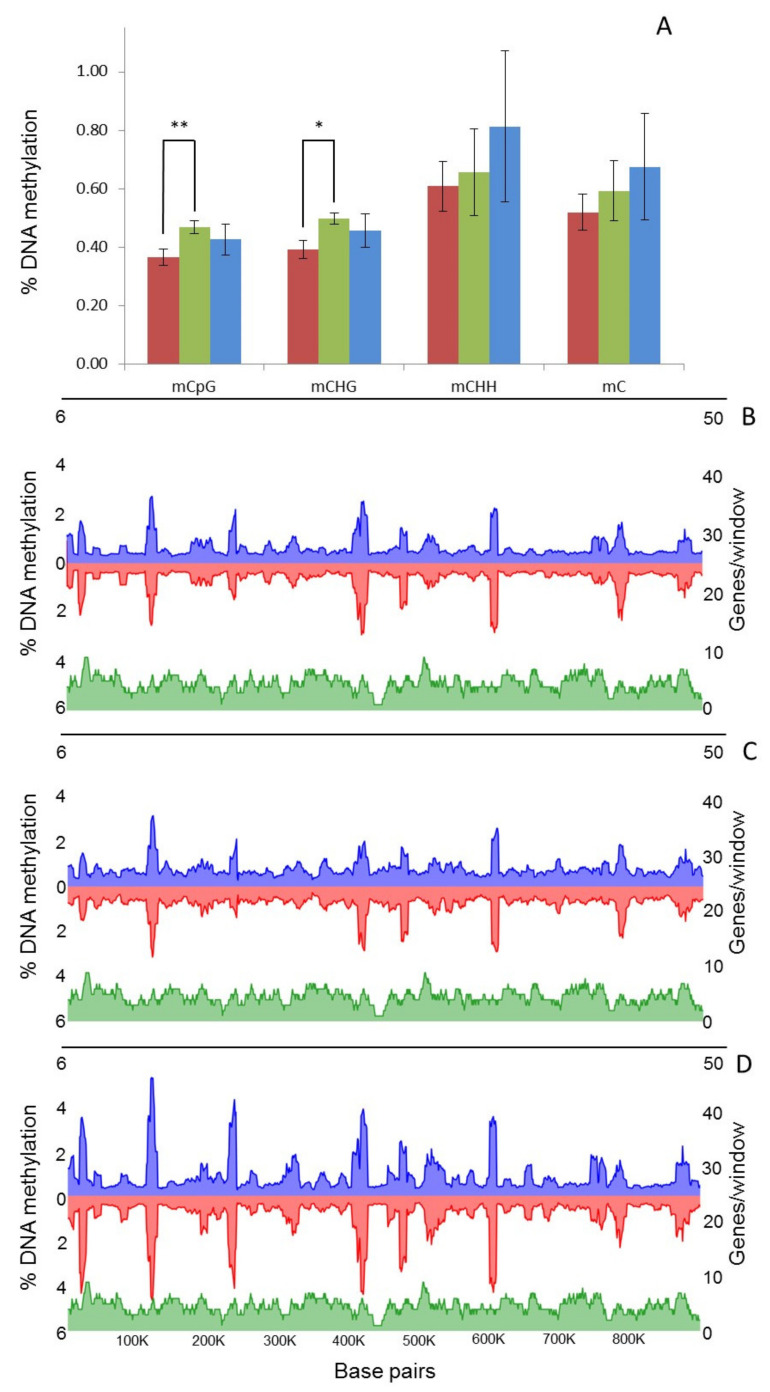
Global changes on genomic distribution and levels of DNA methylation in *B. cinerea*. (**A**) Global average percentage of mCs ((number of mCs/total Cs) × 100) at each time point (T1, red; T8, green; T8P, blue). (**B**–**D**) Methylcytosines (mCs) density from each strand (blue, positive and red, negative strand) across Supercontig 1.1 at each time point (**B** = T1; **C** = T8 and **D** = T8P) was calculated and plotted as the percentage of mCs (as above) in each 10 kb window. * (*t*-test; *p* < 0.05) and ** (*t*-test; *p* < 0.01).

**Figure 4 ijms-23-03034-f004:**
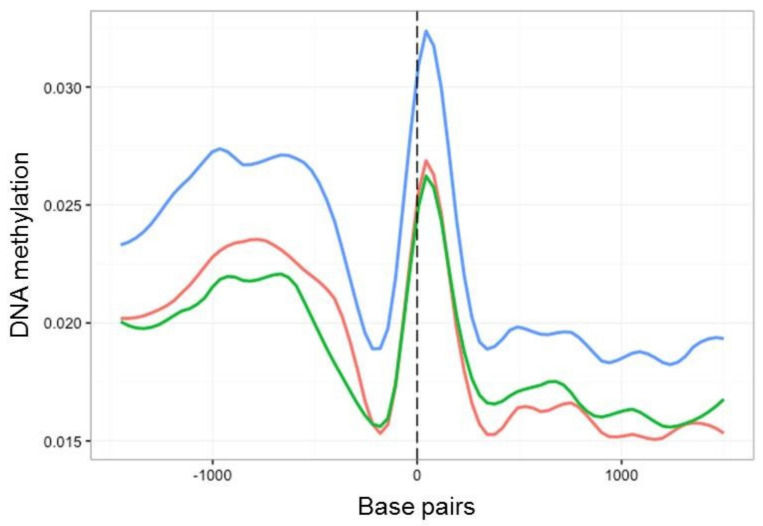
Methylation density across transcription start site (TSS) regions of all *B. cinerea* selected genes. Methylation level (vertical axis) at each time point (T1, red; T8, green; T8P, blue) was identified as the proportion of methylated cytosines against all cytosines in 30 bp windows 1.5 kb before and after the transcription start site.

**Figure 5 ijms-23-03034-f005:**
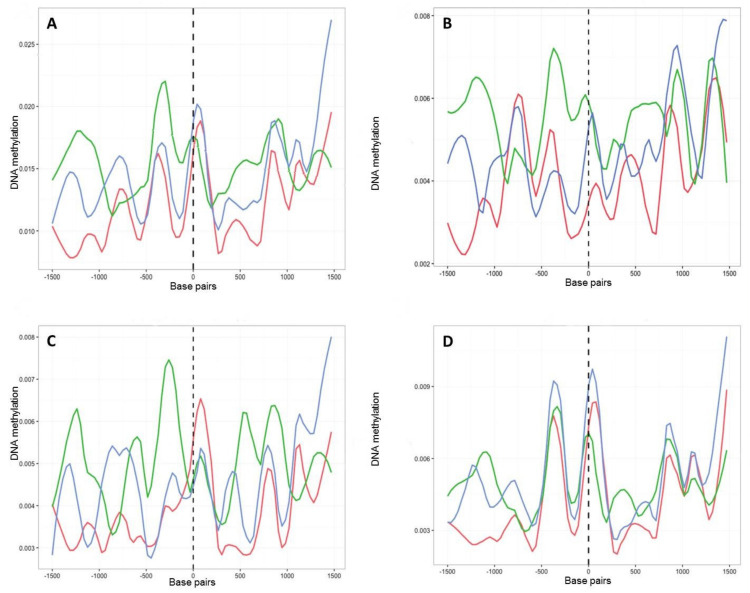
Methylation density across transcription start sites (TSSs) found in 131 genes associated with polysaccharide degradation in *B. cinerea* (27). Methylation level (vertical axis) at each time point (T1, red; T8, green; T8P, blue) was identified as the proportion of methylated cytosines against all cytosines in 30 bp windows 1.5 kb before and after the transcription start site (TSS) (horizontal axis). (**A**) Methylation level for all mCs; (**B**) methylation level for CpG context; (**C**) methylation level for CpHpG context; and (**D**) methylation level for CpHpH context.

**Figure 6 ijms-23-03034-f006:**
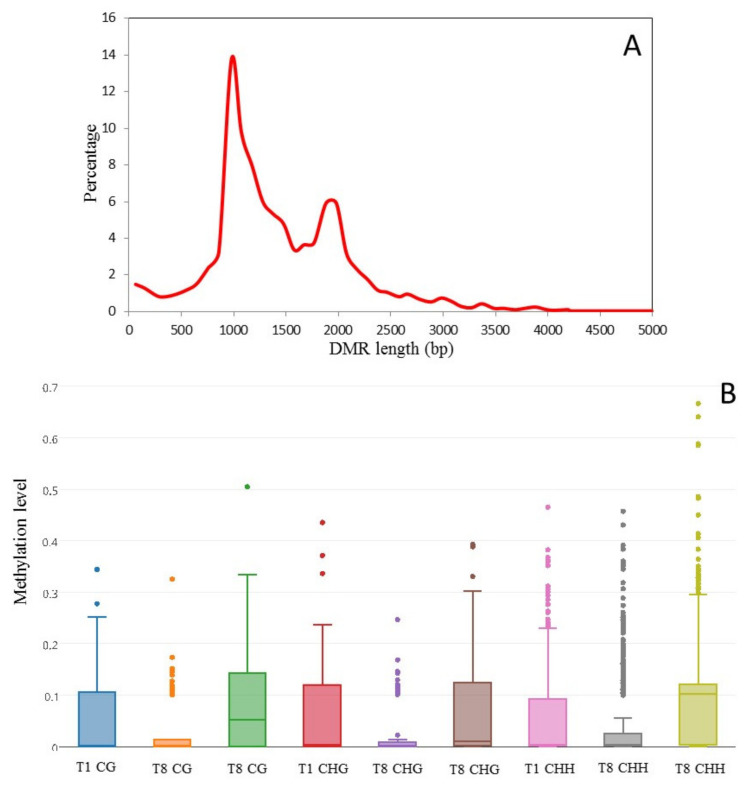
Analysis of in vitro culture-induced DMRs in *Botrytis cinerea***.** (**A**) Length distribution of in vitro culture-induced DMRs (i.e., regions presenting significantly different methylation levels between one sample and the other two samples (FDR < 0.01)) in *Botrytis cinerea.* DRMs were determined using a three-sample Kruskal-Wallis test. Methylation levels were analysed by sliding window analysis using swDMR. (**B**) Methylation level distribution in in-vitro-induced DMRs. Boxplot of 3-sample sliding window differential methylation analysis using swDMR. The boxplots show the distribution of methylation in each context (CG, CHG and CHH).

**Table 1 ijms-23-03034-t001:** *Botrytis cinerea* genes with known function presenting a sequence variant and/or overlapping with in vitro culture-induced DMRs. Columns that Gene Variant and Gene/DMR overlap indicate the number (#) and percentage (%) of genes in each functional group with a genetic variant or overlapping with a differentially methylated region, respectively. Column DMR type indicates how methylation levels changed when comparing all three samples (T1, T8 and T8P). DMRs were grouped according to their changing patterns into recovery (T1 = T8P) and non-recovery (T1< >T8P). Two subgroups were found for recovery (T1 = T8P < T8 (Type 0) and T1 = T8P > T8 (Type 2)) and non-recovery (T1 > T8 = T8P (Type 1a) and T1 = T8 < T8P (Type 1b)). Virul. s.l. indicates virulence genes in abroad sense and includes genes associated with: appressorium formation, virulence in a strict sense (virulence *s.s.*) and CAZyme genes.

Gene Function		Total	Gene Variant		Gene/DMR Overlap		DMR Type				Reference
			#G/D	%G/D	#G/D	%G/D	Recovery		Non- Recovery		
							0	2	1a	1b	
Housekeeping		5	-	-	-	-	-	-	-	-	[25]
Apoptosis		10	2	20	4	40.0	3	1	-	-	[25]
Conidiation		16	-	-	6	37.5	1	2	-	3	[25]
Mating and fruit body development		32	6	18.8	6	18.7	1	4	1	1	[25]
Secondary metabolism		51	4	7.8	14	27.5	9	3	2	3	[25]
Signalling pathways		176	6	3.4	54	30.7	12	17	7	20	[25]
Sclerotium formation		249	7	2.8	67	26.9	26	31	10	13	[25]
**Virul. *s.l.***	Appressorium	12	2	16.7	3	25.0	1	-	-	3	[25]
	Virulence *s.s.*	17	1	5.9	4	23.5	1	1	1	2	[26]
	CAZyme genes	1155	50	4.3	320	27.7	112	132	45	99	[27]
**Total**		**1577**	**68**	**4.3**	**478**	**30.3**	**166**	**191**	**66**	**144**	

**Table 2 ijms-23-03034-t002:** Number of differentially methylated regions (DMRs) between T1, T8 and T8P samples. DRMs (i.e., regions presenting significantly different methylation levels between one sample and the other two samples (FDR < 0.01)) were determined using a three-sample Kruskal-Wallis test. Methylation levels were analysed by sliding window analysis using swDMR. DMRs were determined for all cytosines and for three methylation contexts. DMRs were grouped according to their changing patterns into recovery (T1 = T8P) and non-recovery (T1< > T8P). Two subgroups were found for recovery (T1 = T8P < T8 (Type 0) and T1 = T8P > T8 (Type 2)) and non-recovery (T1 > T8 = T8P (Type 1a) and T1 = T8 < T8P (Type 1b)). Percentage of the total DMRs for each pattern type/sequence context is shown in parenthesis.

	Total	Recovery (%)		Total Recovery (%)	No Recovery	
		Type 0	Type 2		In Vitro Induced (Type 1a)	Plant Induced (Type 1b)
mC	2822	757 (26.82)	860 (30.47)	1617 (57.30)	452 (16.02)	753 (26.68)
CG	70	17 (24.29)	15 (21.43)	32 (45.71)	14 (20.00)	24 (34.29)
CHG	82	14 (17.07)	30 (36.59)	44 (53.66)	15 (18.29)	23 (28.05)
CHH	1248	303 (24.28)	490 (39.26)	793 (63.54)	137 (10.98))	318 (25.48)
Total	4222	1395 (33.04)	1091 (25.84)	2486 (58.88)	618 (14.64)	1118 (26.48)

**Table 3 ijms-23-03034-t003:** *Botrytis cinerea* in vitro induced differentially methylated regions overlapping with genes. DRMs overlapping with genes (i.e., regions presenting significantly different methylation levels between one sample and the other two samples (FDR < 0.01)) were determined using a three-sample Kruskal-Wallis test. Methylation levels were analysed by sliding window analysis using swDMR. DMRs were determined for all cytosines and for three methylation contexts. DMRs were grouped according to the genic region they overlapped with (i.e., promoter, promoter and gene body, promoter, gene body and 3’UTR, gene body and 3′UTR and gene body). (**) Percentage of the total DMRs overlapping with each particular genic region. (*) Percentage of the total number of genes showing a methylation recovery pattern.

	Total Genes	Promoter, Gene Body and 3’UTR (*)	Promoter Only (*)	Promoter and Gene Body (*)	Gene Body and 3’UTR (*)	Gene Body (*)
mC	3055	626 (20.49)	61 (2.00)	1022 (33.45)	923 (30.21)	423 (13.85)
CG	68	3 (4.41)	0 (0.00)	24 (35.29)	21 (30.88)	20 (29.41)
CHG	84	8 (9.52)	0 (0.00)	29 (34.52)	27 (32.14)	20 (23.81)
CHH	1339	248 (18.52)	32 (2.39)	443 (33.08)	413 (30.84)	203 (15.16)
	Recovery genes (**)					
mC	1713 (56.07)	345 (20.14)	43 (2.51)	545 (31.82)	537 (31.35)	243 (14.19)
CG	32 (47.06)	2 (6.25)	0 (0.00)	11 (34.38)	9 (28.13)	10 (31.25)
CHG	46 (54.76)	4 (8.70)	1 (2.17)	13 (28.26)	16 (34.78)	12 (26.09)
CHH	863 (64.45)	175 (20.28)	21 (2.43)	286 (33.14)	252 (29.20)	129 (14.95)

## Data Availability

The data presented in this study is at the Sequence Read Archive (PRJNA808095).

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
