# Peer review of "Botrytis cinerea Loss and Restoration of Virulence during In Vitro Culture Follows Flux in Global DNA Methylation"

_ijms, 2022, doi:10.3390/ijms23063034_

Round 1

Reviewer 1 Report

The authors address an interesting question regarding transient (partial) loss of virulence after prolonged culture on artificial medium. The goal of this work was to test whether this is due to mutation of to epigenetic phenomena associated with reversible DNA methylation. For this, they performed 8 transfers over a period of 32 weeks, followed by infection of the plant and recultivation of the spores produced on the infected tissue. Changes in the methylation state were scored using Methylation-sensitive amplified polymorphisms (MSAP).

I don’t think that the authors used  the best experimental setup. Why did they only 8 transfers every 4 weeks, and not a much higher number of transfers performed every 10-14 days (Botrytis is able to fully spread and sporulate on a plate within 7-10 days) to see clear effects? So, the cultures were left in a grown up stage (plates sealed with parafilm or slowly drying out? No description about the culture conditions except for a reference [42] which leads no nowhere!), and it is not clear whether the effects of serial cultures or of long-term storage under undefined conditions is analysed here. Also, infection phenotypes on Arabidopsis are difficult to score quantiatively (as shown in Fig. 1, in which a very rough scoring method was used), a more sensitive tissue such as tomato or Phaseolus leaves would have given much more precise values.

I don’t really understand the results of the genome sequencing. From all what we know about stability of fungal genomes, it is highly unlikely that 8 transfers (or ‘8-months culturd’) lead to 2331 mutations (44% small indels, 56% SNPs). For comparison, we have recently performed genome sequencing with B. cinerea multiple mutants generated by CRISPR in up to 7 successive rounds of transformation, single spore isolation, transfers and retransformation (https://www.biorxiv.org/content/biorxiv/early/2021/08/22/2021.08.21.457223.full.pdf), and found by genome sequencing (unpublished) less than 10 mutations, consistent with the work of others who found a great stability of the fungal genome even after 400 days of in vitro growth (Kohn et al., 2008,  Fungal Genet. Biol. 45: 613). I would like to see more of the original data in a supplement, which would allow to critically evaluate the results.

In the introduction, the authors almost exclusively cite old and difficult to access publication. For example, the reference to the sentence ‘Degenerated cultures have been reported in a wide range of pathogenic fungi [5]’ is citing an old book which is only dealing with entomopathogens, citations 6-9 don’t seem to address the topic of this manuscript, and citations 12-15 are not up-to-date of the dynamic field of epigenetics, and should focus on recent literature with fungal pathogens, for example Chang et al. (2020) ‘ Degenerated Virulence and Irregular Development of Fusarium oxysporum f. sp. niveum Induced by Successive Subculture. J Fungi (Basel). 2020 Dec; 6(4): 382.’

Other references don’t fit to the text, e.g.  ref. 2 to ‘Recent years have seen a dramatic increase in the depth of understanding of how epigenetic control mechanisms operate during plant/pathogen interactions [2,22].’

Reviewer 2 Report

James Breen and colleagues present an article on the correlation between virulence and altered DNA methylation patterns in the pathogenic fungus Botrytis cinerea, monitored at genome wide level and for longer time periods. In general the article is well written, contains interesting observations and discusses possible scenarios and speculative aspects in an appropriate manner. Overall the presented experiments and the provided statistical analyses appear to be well conducted and correctly interpreted.

In my opinion, the Abstract would benefit from some shortening. Perhaps lines 18-22 could be omitted.

Given that the discussion section is quite extensively written, the authors should try to keep any repetition of results to a minimum.

Authors may want to consider improving the quality of Figure 1, perhaps by adding color bars in the diagrams, and adding a petri dish with a growth test or a microscopic image of Botrytis cinerea.

Line 73: please correct the genus name.

Line 160: please correct “rasther”.

Line 177: please correct “senso lato”.

Line 222: please italicise B. cinerea.

Line 401: please delete “rich”.

Lines 457-458: please delete either “showing” or “containing”.

Line 478: please rephrase.
